Architecture of the sperm whale forehead facilitates ramming combat

Panagiotopoulou Olga 1 o.panagiotopoulou@uq.edu.au
Spyridis Panagiotis 2 3
Mehari Abraha Hyab 1
Carrier David R. 4
Pataky Todd C. 5
1 Faculty of Medicine and Biomedical Sciences, Moving Morphology & Functional Mechanics Laboratory, School of Biomedical Sciences, The University of Queensland , Brisbane , Australia
2 Polytropos Ltd , London , United Kingdom
3 Department of Civil Engineering and Natural Hazards, University of Natural Resources and Life Sciences , Vienna , Austria
4 Department of Biology, University of Utah , Salt Lake City , Utah, United States of America
5 Institute for Fiber Engineering, Department of Bioengineering, Shinshu University , Ueda, Nagano , Japan
Farke Andrew
Electronic publication date: 2016 Apr 5
Publication date: 2016
Volume: 4
Electronic Location ID: e1895
Received 2015 Dec 16; Accepted 2016 Mar 14
Copyright: ©2016 Panagiotopoulou et al.
Copyright year: 2016
Copyright holder: Panagiotopoulou et al.
License: This is an open access article distributed under the terms of the Creative Commons Attribution License, which permits unrestricted use, distribution, reproduction and adaptation in any medium and for any purpose provided that it is properly attributed. For attribution, the original author(s), title, publication source (PeerJ) and either DOI or URL of the article must be cited.
License URL: https://creativecommons.org/licenses/by/4.0/

Keywords: Sperm whale, Spermaceti junk, Ramming impact, Finite element analysis, Probabilistic simulation, Connective tissue partitions

Funding: The authors received no funding for this work.

==============================
Herman Melville’s novel Moby Dick was inspired by historical instances in which large sperm whales (Physeter macrocephalus L.) sank 19th century whaling ships by ramming them with their foreheads. The immense forehead of sperm whales is possibly the largest, and one of the strangest, anatomical structures in the animal kingdom. It contains two large oil-filled compartments, known as the “spermaceti organ” and “junk,” that constitute up to one-quarter of body mass and extend one-third of the total length of the whale. Recognized as playing an important role in echolocation, previous studies have also attributed the complex structural configuration of the spermaceti organ and junk to acoustic sexual selection, acoustic prey debilitation, buoyancy control, and aggressive ramming. Of these additional suggested functions, ramming remains the most controversial, and the potential mechanical roles of the structural components of the spermaceti organ and junk in ramming remain untested. Here we explore the aggressive ramming hypothesis using a novel combination of structural engineering principles and probabilistic simulation to determine if the unique structure of the junk significantly reduces stress in the skull during quasi-static impact. Our analyses indicate that the connective tissue partitions in the junk reduce von Mises stresses across the skull and that the load-redistribution functionality of the former is insensitive to moderate variation in tissue material parameters, the thickness of the partitions, and variations in the location and angle of the applied load. Absence of the connective tissue partitions increases skull stresses, particularly in the rostral aspect of the upper jaw, further hinting of the important role the architecture of the junk may play in ramming events. Our study also found that impact loads on the spermaceti organ generate lower skull stresses than an impact on the junk. Nevertheless, whilst an impact on the spermaceti organ would reduce skull stresses, it would also cause high compressive stresses on the anterior aspect of the organ and the connective tissue case, possibly making these structures more prone to failure. This outcome, coupled with the facts that the spermaceti organ houses sensitive and essential sonar producing structures and the rostral portion of junk, rather than the spermaceti organ, is frequently a site of significant scarring in mature males suggest that whales avoid impact with the spermaceti organ. Although the unique structure of the junk certainly serves multiple functions, our results are consistent with the hypothesis that the structure also evolved to function as a massive battering ram during male-male competition.

Introduction

The sperm whale (Physeter macrocephalus L.) is unique in having a massively expanded forehead that is highly sexually dimorphic, being much larger and extending up to a meter and a half beyond the anterior tip of the jaws in mature males (Berzin, 1972; Cranford, 1999). Internally the forehead is composed of two large oil-filled sacs, stacked one on top of the other, known as the dorsal spermaceti organ (or spermaceti case) and the junk (Fig. 1). These sacs extend for one-third of the total length of the whale and can constitute more than one-quarter of the whale’s mass (Berzin, 1972; Clarke, 1978). The oil contained in the upper sac (spermaceti organ) was a primary target of the whaling industry of the early 19th century. At the same time, the forehead of sperm whales was considered by whalers to be a battering ram that the whales sometimes used to attack and sink oak whaling ships of up to 238 tons (Chase, 1821; Starbuck, 1878; Philbrick, 2000).

The lower sac (the junk) is derived from the odontocete melon (Heyning & Mead, 1990) and is organized into sections by transverse partitions of connective tissue that contain waxy oil (Clarke, 1978) (Fig. 1). The connective tissue partitions embedded in the junk are widest about 10–25% of the length from the anterior end and the sections are narrow ventrally and broad dorsally (Clarke, 1978). The partitions become thinner progressively towards the posterior aspect of the junk until they are totally replaced by a mixture of oil and wax. The oil and connective tissue partitions of the junk are enclosed in a fibrous connective tissue case which sits in a trough formed by the upper jaw (Clarke, 1978).

The function of the spermaceti organ and the junk in adding directionality and amplitude to sonar clicks is relatively well studied and accepted (Møhl et al., 2000; Møhl, 2001; Møhl et al., 2003a; Møhl et al., 2003b; Huggenberger, André & Oelschläger, 2014). Previous studies have also suggested that the unique structural configuration of the sperm whale forehead is functionally related to acoustic sexual selection (Cranford, 1999), acoustic prey debilitation (Norris & Møhl, 1983), communication (Madsen, Wahlberg & Møhl, 2002) and buoyancy control (Clarke, 1970). Although all of these functional hypotheses are plausible, they cannot explain how the forehead of sperm whales may function as a battering ram capable of sinking ships that are four to five times the mass of the whale.

Figure 1 Schematic representation of sperm whale head structure. Image courtesy of Ali Nabavizadeh.

The ramming hypothesis was originally proposed by whalers following the sinking of at least 2 whaling ships, the Essex in 1821 and the Ann Alexander in 1851 (Chase, 1821; Starbuck, 1878; Philbrick, 2000; Sawtell, 1962). Based on these incidents, researchers have recently suggested that the forehead of a swimming sperm whale may possess sufficient momentum to injure an opponent when used as a battering ram, and may at the same time absorb energy to protect the brain and skull of the attacking whale allowing mature males to use their foreheads as battering rams in male–male contests over harems of females (Carrier, Deban & Otterstrom, 2002).

The ramming hypothesis remains highly controversial because (1) the structures that generate sound, the distal sac and monkey lips of the right nasal passage, are located at the rostral end of spermaceti organ (Fig. 1) and are therefore assumed to be in harm’s way in a ramming event (Huggenberger, André & Oelschläger, 2014), and (2) ramming episodes have not been observed by scientists who study the behavior of sperm whales. Although the monkey lips do reside at the front end of the spermaceti organ, these structures are located well above and to the right of the rostral end of the junk (Huggenberger, André & Oelschläger, 2014), and it is the junk, not the spermaceti organ, that has been suggested to function as a battering ram during aggressive encounters (Chase, 1821; Carrier, Deban & Otterstrom, 2002). As far as we know, the scientific literature does not include observations of sperm whale ramming behavior, yet there is one documented observation of male-male ramming that we report here (Supplemental Information 1). On January 30, 1997, a wildlife pilot, while flying over the Gulf of California, watched two mature males swim directly toward each other, from an initial observed distance of approximately 6.4 km, at an estimated average swimming speed of 17 km/h and collide forehead-to-forehead. Shortly before impact both whales, which had been swimming at the surface, “shallow dove” so that the impact occurred below the surface of the water. This ramming event occurred a few miles north of a group of approximately of 50 females. This observation plus reports of ramming attacks on 19th century whaling ships suggest that sperm whales may indeed engage in ramming contests. If these ramming contests generally occur at a shallow depth, they may be much more common than whale biologists realize because a human observer would have to be located well above the surface of the water to watch it happen.

Another reason to consider the ramming hypothesis is the extreme body size sexual dimorphism of sperm whales. This species is the most sexually dimorphic of all cetaceans, with mature males being 3-times bigger than mature females (Whitehead, 2003). Among mammals, body size sexual dimorphism is generally greatest in polygynous species in which males compete through fighting and the threat of fighting (Clutton-Brock & Harvey, 1977; Parker, 1983; Andersson, 1994). Additionally, because sexual dimorphism is often greatest in those characters that enhance a male’s capacity to dominate other males (Clutton-Brock & Harvey, 1977; Hamilton, 1979; Clutton-Brock, Albon & Harvey, 1980; Parker, 1983; Jarman, 1983; Andersson, 1994), the observation that head length is the most dimorphic anatomical feature (Nishiwaki, Ohsumi & Maeda, 1963) is consistent with the head being a weapon important to male-male competition.

This paper addresses the battering-ram hypothesis using finite element analysis and probabilistic simulation. Our main objective was to determine if the connective tissue partitions of the junk have potential to reduce stress in the bones of the skull during ramming impact. We predict that the vertically oriented connective tissue partitions within the junk can dissipate load through tension during posteriorly directed compressive loading of the forehead. Bone stress reduction would be particularly important on the anterior aspect of the skull (i.e., upper jaw) that would otherwise be most vulnerable to potential tissue damage.

Materials and Methods

Finite element analysis (FEA) is a numerical technique well entrenched in comparative biomechanics as a tool to assess the mechanical architecture of anatomical tissues and to better comprehend the complex interaction of their form–function relationships. Nevertheless, FEA accuracy is dependent on a variety of factors and its reproducibility is often obscured in scientific publications due to both public unavailability of the underlying models and the lack of standard reporting guidelines (Erdemir et al., 2012). To mitigate these problems we here describe our methods in accordance with biomechanical FEA reporting guidelines (Erdemir et al., 2012) and we also make all raw data and FE models available for public use (Panagiotopoulou et al., 2016).

Model identification. Our study utilized three FE models to study the effect of the connective tissue partitions on the reduction of bone stresses in quasi-static loading of the sperm whale head (Fig. 2).

Figure 2 Finite element models.

Young’s moduli for the connective tissue partitions (blue), spermaceti organ (yellow) and skull (red) were, 2 GPa, 1 GPa and, 14.8 GPa, respectively. Models A, B and C have twelve, six and zero connective tissue partitions, respectively.

Model name. Sperm Whale Head Model A Generic Base consisted of twelve connective tissue partitions embedded in the spermaceti tissue of junk.

Sperm Whale Head Model B Half Partitions had reduced number (six) of connective tissue partitions.

Sperm Whale Head Model C No Partitions had no connective tissue partitions.

Model keywords. Sperm whale skull, quasi-static impact.

Version. 0.1 (Panagiotopoulou et al., 2016)

Physiological domain. No segmental motion, evenly distributed anterior surface loading, small deformations of hard and soft tissue.

Mechanical domain. All models were static and linear elastic.

Structure of interest. The biological structure under investigation was the sperm whale upper jaw (skull).

Demographics. Adult male sperm whale (Physeter macrocephalus L.).

State of represented organism. in vitro.

Disease state. Healthy.

Spatial scale. Within a volume of (length 5.3 m × height 1.6 m × width 0.1 m).

Time scale. Not applicable (quasi-static analysis).

Primary utility. To provide mechanical insight into a physiological process.

Secondary utility. First model of sperm whale skull mechanics.

Primary highlight. To elucidate the possible mechanically protective role of the vertical connective tissue partitions within the sperm whale skull.

Secondary highlight. Not applicable.

Primary limitation. Linear isotropic and homogeneous materials.

Due to lack of experimental data on the elasticity of the sperm whale head tissues, anisotropy and heterogeneity, as well as environmental and time dependencies could not be modeled in this study. Thus, isotropy, homogeneity and linear elasticity were assumed and the material properties assigned to each tissue were the closest estimations based on published values of tissues similar to those of interest (Rho, Ashman & Turner, 1993; Shahar et al., 2007). A biologically unrealistic consequence of this assumption was that the dorsal horizontal components of connective tissue partitions provided resistance to compression in the model. To this end we assigned a Young’s modulus (E) value of 14.8 GPa and Poisson (v) value of 0.1 for the skull; E = 2 GPa and v = 0.2 for the connective tissue partitions; and E = 1 GPa and v = 0.49 for the oil/wax mixture enclosed within the spermaceti organ and the junk (Fig. 2) (Rho, Ashman & Turner, 1993; Shahar et al., 2007). Nevertheless, our study was comparative and such an assumption likely created a constant error across all models. Additionally, uncertainties due to material variations had been handled through numerical statistic elaboration of the models. Lastly, the basic mechanism of skull-stress reduction we described was independent of model realism.

Figure 3 Loads and constraints assigned to all FE models.

A force of 764 kN was applied to the anterior surface of the head (1). Motion was constrained at the posterior surface (2) in all directions.

Secondary limitation. Simplifications in the model geometry (see below), static simulation.

Reference to publications. Clarke (1970) and Clarke (1978); no explicit mechanical model described.

Model structure

Loading and boundary conditions. We used a static force of 764 kN (Fig. 3) distributed evenly over the junk of the sperm whale nose as a simplified model of ramming force. Our analysis was primarily concerned with loads applied to the junk instead of the spermaceti organ because the scars on mature males are confined primarily to the front end of the junk, rather than the spermaceti organ per se (Carrier, Deban & Otterstrom, 2002, Fig. 7). Nevertheless, to ensure that the location of the impact force did not influence the validity of our conclusions, we performed a sensitivity analysis by applying (Fig. 3) an acute angle of 20.56°(collinear with the model’s anterior part of the connective tissue) to the impact force. We also varied the application point of the impact force, applying the load to the spermaceti organ, and the superior, middle and inferior aspects of the junk respectively for all models (Fig. S1). The magnitude of the applied force per surface area is shown in Table S1. We calculated the applied force by assuming that each of the two colliding whales were traveling at an intermediate speed of 6.26 ms−1 (Aoki et al., 2007) had masses of 39,000 kg, and decelerated over a distance of 1 m upon impact. The deceleration distance was based on the length of the junk that extended beyond the tip of the skull. Boundary conditions included no-displacement constraints on all external nodes on the posterior surface of the skull.

Primary output variables. von Mises stress.

Source of anatomy. To test our hypothesis, we developed FE models based partially on previously published structural properties and schematic configurations of male sperm whale adult cadavers. Due to the inaccessibility of sperm whale cadaveric species, the report by Clarke (1978) was the most detailed hitherto available and encompassed skeletal and soft tissues such as the connective tissue partitions, and the oil cases of the spermaceti organ and the junk. To calculate the dimensions of the various structural components of the model, we scaled the anatomical elements shown in Fig. 1 of Clarke (1978) to a total spermaceti organ length of 5 m (Clarke, 1970). For modelling purposes and due to the unclear description of the individual connective tissue partitions thickness, we assumed a universal thickness of 0.05 m and 0.150 m for all connective tissue partitions and the junk compartments between the connective tissue partitions respectively (Fig. 2). In order to ensure that the modeling simplification with regards to the thickness of the connective tissue partitions did not influence our overall conclusions and were biologically sound, we conducted a sensitivity analysis and compared the von Mises skull stresses of Model A, against a modification with connective partitions thickness reported by Clarke (1978) and listed in Table S2.

Model A, representative of the sperm whale head, consisted of the upper sack or spermaceti organ; the lower sack or junk; the connective tissue partitions with a universal thickness of 0.05 m and 0.150 m and their subsequent connective tissue case enclosed in the junk and the upper jaw (Fig. 2). Model A2 was identical to Model A but the connective tissue partitions were assigned different thickness (Table S2).

We compared Model A against two modified models (Models B and C) to assess the mechanical function of the spermaceti organ (Fig. 2). Model B had fewer connective tissue partitions than Model A. Model C lacked the connective tissue partitions altogether (Fig. 2).

The skin and the blubber were discarded from the modeling process due to their negligible thickness and stiffness.

The FE mesh assembly of all models consisted of solid continuum linear tetrahedral elements (type “C3D4” in the Abaqus Library; Simulia-Dassault Systemes, Waltham, MA, USA). Each model contained approximately 42,000–48,000 nodes and 220,000–260,000 elements. Model A had 257,542 elements in total (28,009 for the upper jaw; 65,588 for the spermaceti case; 91,272 for the junk; and 72,673 for the connective tissue partitions). Model B had 242,509 elements in total (27,896 for the upper jaw; 65,467 for the spermaceti case; 93,482 for the junk; and 55,664 for the connective tissue partitions). Model C had 227,925 elements in total (28,137 for the upper jaw; 65,519 for the spermaceti case; 134,269 for the junk. The nominal element size was 50 mm (0.05 m), and the actual elements sizes across the model varied approximately from 15 to 85 mm.

Reference configuration. The Abaqus default x (cranial-caudal), y (medial-lateral), z (vertical) coordinate system was used.

Simulation structure

Name of simulation software. Abaqus/CAE (Simulia-Dassault Systemes).

Version of simulation software. 6.12

Solution strategy. Abaqus/Standard implicit direct static solver. Minimum and maximum increments set to 1.000E–05 and 1 respectively.

Numerical algorithms. Full Newton default iterations.

Convergence criteria. Default convergence tolerances of the simulation software were used. We interpreted stress differences amongst our models using a Monte Carlo simulation. A total of 1,000 Monte Carlo iterations were run for each of the three models, varying the three materials’ stiffness values randomly with a standard deviation of 10%, and von Mises stress distributions were stored for each iteration (Supplemental Information). This resulted in a population of 1,000 random individuals which represented the population of interest, under an assumption of 10% error in each of the material parameters. The latter is an essential approach in cases when the assigned material properties are based on generalized published values and not on experimental analysis of the tissues of interest.

For each population pair (i.e., Model B vs. Model A, Model C vs. Model B and Model C vs. Model A), the following statistic was calculated for each element: zi=si¯B−si¯A12σiA+σiB

where i indexes the elements, A and B represent models, si¯ represents mean elemental von Mises stress and σi represents elemental standard deviation. Since the input variance nonlinearly maps to elemental variance, the z distribution is a non-trivial function of mean group differences.

Next, the “significance” of the z distribution was assessed using paired non-parametric permutation tests (one for each model pair). A total of 10,000 label permutations were applied to each model pair, yielding a non-parametric distribution of the z statistic at each element. The 99th percentile of that distribution was taken as the “significance” threshold. In other words, if an element’s z value survived that threshold, it would suggest that 99% of all randomly labelled individuals would yield a z value less than that observed in the original labellings, and thus that threshold-surviving elements represented true population differences at alpha =0.01 under the assumption of 10% true population material parameter variance.

Validation. Validation of the FE models against experimental ex vivo data was not feasible due to size and accessibility constraints. Nevertheless our study is comparative and conclusions are fundamentally mechanical rather than empirical.

Availability. Not yet public.

Results

In all our FE models the highest concentration of von Mises stresses occurred in the most anterior aspect of the skull (Figs. 4 and 5 and Table 1). The anterior connective tissue partitions within the junk were subjected to higher tensile loading than the posterior portions (Fig. 6). Tension in the connective tissue partitions redistributed compressive stresses across the skull (Models A and B) and the absence of the partitions (Model C) raised anterior skull stresses (Figs. 4 and 5 and Table 1).

Figure 4 Von Mises stress distribution results.

Figure 5 Region definitions (blue vertical bars).

Table 1 Maximum (Max.), mean and minimum (Min.) percentage increase of the regional (Fig. 5) von Mises stress values (Pa) between Models A and C and Models B and C.

% increase	
	1	2	3	4	5	
	Max.	Mean	Min.	Max.	Mean	Min.	Max.	Mean	Min.	Max.	Mean	Min.	Max.	Mean	Min.	
Model A– Model C	42.9	45.7	3.6	25	24.1	9.4	−8.8	6.1	60.8	−6.5	−0.9	−27.3	1.4	−0.3	−62.8	
Model B– Model C	10.1	15.5	12.8	−7.3	4.1	35.7	0.6	0.8	−6.9	−6.7	−1.4	−27.4	1.4	−0.2	−59.6	

Figure 6 Maximum principal stress distributions across the connective tissue partitions.

Positive and negative stresses indicate areas of tension and compression respectively.

Figure 7 Probabilistic FEA simulation.

Z statistic distributions depicting mean elemental von Mises stress differences divided by elemental standard deviation under an assumed population material stiffness variance of 10%. Data are thresholded at alpha = 0.01.

A reduced number of partitions (Model B) did reduce stresses in the anterior skull, but stress reduction was not as effective as Model A (Figs. 4 and 5 and Table 1). The skull stress difference distributions resulting from Monte Carlo simulations suggest that our main finding regarding the load-redistribution functionality of the connective tissue is insensitive to relatively large changes in both material parameters (Fig. 7) and, indirectly, load magnitudes.

Our sensitivity analysis of the load location further supported our findings that the connective tissue partitions reduce stresses on the anterior skull; however, stress magnitudes on the skull and the connective tissue case were sensitive to variations of the load location (Figs. S2–S6). In all comparisons, the highest stresses were found to occur in Model C, which lacked the connective tissue partitions and the lowest stresses occurred in Model A, with the twelve connective tissue partitions (Figs. S2–S6). Load application on the spermaceti organ generated lower skull stresses than an impact load on the junk (Figs. S2–S6), yet it increased stress concentrations on the superior aspect of the connective tissue case and the rostral end of the spermaceti organ, where the sound generator of the sonar system (monkey lips) is housed (Fig. 8). Although, impact load on the junk created higher skulls stresses (Figs. S2–S6), it created lower connective case stresses (Fig. 8) than an impact load on the spermaceti organ. In addition, as the loading site on the anterior junk moved inferiorly, the connective case stresses were reduced, whereas the skull stresses increased and the connective tissue partitions underwent increased tension. From the top of the anterior junk to the bottom, variations in the angle of the applied force on the junk increased skull stresses on the posterior aspect of the skull in all models, yet the model without partitions (Model C) showed the highest stress concentration across the whole skull and thus the least resistance to bending (Fig. S6).

Figure 8 Von Mises stress distribution results following variations in the topological application of the impact force.

(A) Impact force on the spermaceti organ; (B) Impact force on the superior aspect of the spermaceti junk; (C) Impact force on the mid spermaceti junk; (D) Impact force on the inferior aspect of the spermaceti junk; (E) Impact force on the entire anterior aspect of the spermaceti junk. Warm (red) and cold (blue) colors show higher and lower von Mises stresses respectively.

The results of our sensitivity analysis on the partition thickness showed that the skull stresses are insensitive to variations in the partitions’ dimensions (Fig. S7). Model A with the simplified partitions (Fig. S7) showed slightly decreased stresses on the mid-anterior aspect of the skull than Model A2, with partition thickness according to Clark’s observations. Nevertheless, the skull von Mises differences were minimal and do not influence the general comparisons between Models A (with twelve connective tissue partitions) and Model C (with no connective tissue partitions). This raises confidence that the model simplifications on the partition design for Models A and B did not compromise the biological results of our study.

From solely an engineering perspective, Model A with the higher number of connective tissue partitions had higher overall stiffness and potentially lower stress response than Model B with fewer partitions and Model C with no partitions. To test the potential effect of system stiffness on our model comparisons, we measured the reaction forces at the back of the skull (location of the constraints) in all models, after applying a uniform horizontal displacement 0.1 m at the load application location on the junk. The results gave a reaction forces of 9.56E + 06, 9.16E + 06 and 7.8E + 06N for Models A, B and C respectively (Supplemental Information 2 and 3). This suggested that whilst the stiffness between Models A and B was quite similar, Model C was substantially different. To ensure that the comparisons between the models reflected a true effect of architectural design rather than solely being due to the changes in the system stiffness, we varied the stiffness of Models B and C by changing the Young’s modulus of the connective tissue until we obtained similar reaction force for all models (Fig. S8). Comparisons of von Mises stress between Models A, B, and C with equivalent stiffnesses further supported our findings that Model C increases stress on the anterior aspect of the skull (Fig. S8).

Discussion

Our findings suggest that the connective tissue partitions of the junk may reduce impact stresses and thus potentially function as a protective mechanism during ramming. This load-redistribution mechanism of the connective tissue is insensitive to changes in material parameters, load locations and variations in the partitions’ thickness. The mechanism of skull stress reduction appears to be connective partition tension; as the junk is compressed upon impact, the oil between the partitions is displayed vertically and laterally, placing the connective tissue partitions into tension (Fig. 6). This connective tissue tension allows the total compressive bone load to be shared over a greater volume (Figs. 4–7). While our static simulations do not quantify dynamic effects of the connective partitions including energy dissipation, our results suggest that connective partition tension would in fact dissipate energy in dynamic impacts because dynamic loading, like the current quasi-static loading, would be distributed over a broader region of the skull. Additionally, during a dynamic impact soft tissues within the skull would displace, and the connective tissue partitions would limit this displacement. The connective tissue might therefore protect both bone and soft tissue from injury. Absence of the partitions increased stresses by 45%, concentrated on the most anterior aspect of the skull, making the skull more prone to tissue failure (Table 1 and Figs. 4–7).

From an engineering perspective one could argue that the mechanical effect of the connective tissue partitions in minimizing skull stresses might be due to variations in the system stiffness rather than the actual architecture of the junk. Our stiffness sensitivity analysis, however, confirmed that the connective tissue partitions are insensitive to system stiffness. Thus, the architecture and anatomical configuration of the connective tissue partitions assist in reducing impact stresses, particularly on the anterior aspect of the skull.

Our modeling indicates that an impact load applied to the spermaceti organ generates lower skull stresses than the impact loads on the junk. Based on this finding and the fact that the rostral end of spermaceti organ often extends anteriorly beyond the junk, it would be expected for sperm whales to use the spermaceti organ during head-butting incidents. However, scarring on the forehead of adult male sperm whales is largely confined to the rostral end of the junk, rather than the spermaceti organ (Carrier, Deban & Otterstrom, 2002, Fig. 7), suggesting that sperm whales avoid contact with the spermaceti organ and use the junk in ramming events. Behaviorally, confining impact to the junk makes sense given the lack of structural reinforcement of the spermaceti organ and the presumably delicate sound generator of the sonar system housed in the anterior spermaceti organ that could potentially be injured in a ramming event (Huggenberger, André & Oelschläger, 2014). In addition, whilst impacts on the spermaceti organ appear more effective in reducing skulls stresses than impacts on the junk in our static model, the former cause an increased concentration of compressive stresses on the spermaceti organ and also on the connective tissue case. Because connective tissue cannot offer resistance to compressive forces, compressive forces applied to the case would tend to cause its collapse resulting in (1) an ineffective ram applied to the opponent whale and (2) secondary impact of the junk. Additionally, compressive collapse of the connective tissue case would not only endanger the sonar apparatus but would ultimately greatly increase skull stresses.

Our findings appear to provide an explanation for previous observations that, in real whales, the partitions become progressively thinner posteriorly until they are replaced by a mixture of oil and wax (Clarke, 1978). The anterior thicker partitions are subjected to the greatest tensile loading (Fig. 6) and, if the battering ram hypothesis is correct, they likely play the biggest role in skull stress reduction in the face of posteriorly-directed impact forces.

The connective tissue partitions of the junk are acquired traits that likely facilitate a variety of functions. In addition to echolocation (Madsen, Wahlberg & Møhl, 2002), the partitions may play an important role in the dissipation of stresses during ramming combat to protect the skull and brain. This “mechanical advantage” is a trait that is likely related to selection on male-male aggressive behavior. Such developmentally non-independent morphological features of the junk are an example of how a derived structure, such as the connective tissue partitions, facilitates evolutionary modifications while maintaining functional integrity (Wagner & Altenberg, 1996).

Our results are not directly relevant to the behavioral strategies behind ramming impacts; however, our findings are consistent with the hypothesis proposed in 1821 by Owen Chase (Chase, 1821). Following the sinking of the Essex whaling ship, Owen Chase hypothesized that sperm whales not only use their immense and elaborately complex foreheads as battering rams when fighting, but also that “the whale’s head is admirably designed for this mode of attack.” The prevalence of head-butting in sperm whales is not well documented. However, ramming is a basal behavior for Bovidae (Farke, 2008; Alvarez, 1990) and Cetacea (Carrier, Deban & Otterstrom, 2002), including humpback whales (Baker & Herman, 1984), bottle-nosed whales (Gowans & Rendell, 1999), narwhales (Silverman & Dunbar, 1980), long-finned pilot whales (Reilly & Shane, 1986) and killer whales (Goley & Straley, 1994). Based on these reports, it has previously been hypothesized that the spermaceti organ of male sperm whales may function as a weapon and is more developed in males due to sexual selection (Carrier, Deban & Otterstrom, 2002). If this is true, then males may be exposed to increased stresses during head-butting ramming and as such necessitate additional support via a dramatically increased and more structurally robust melon.

Our study illustrates how structural engineering principles and probabilistic simulation can be used to address hypotheses of mechanical function in biological systems that are too big or inaccessible to be studied directly. We anticipate our study will stimulate future research aimed at unraveling the mechanical function of the head during aggressive head-butting and ramming in other species such as the Hippopotamus (Kingdon, 1979) in which head-butting aggressive behavior is common but remains unsimulated.

Supplemental Information

Supplemental Information 1 Observation of ramming sperm whales

Report by Sandra Lanham on ramming sperm whales

Click here for additional data file.

Supplemental Information 2 Reaction force estimations of Models A, B and C

Click here for additional data file.

Supplemental Information 3 Reaction force raw data following stiffness optimization for Models A, B and C

Click here for additional data file.

Figure S1 Location of the force assigned to all FEMs to evaluate the sensitivity of the skull stresses on variations of the topological application of the impact force

(A) Impact force on the spermaceti organ; (B) Impact force on the superior aspect of the spermaceti junk; (C) Impact force on the mid spermaceti junk; (D) Impact force on the inferior aspect of the spermaceti junk.

Click here for additional data file.

Figure S2 Von Mises stress distribution results when the impact force was applied on the spermaceti organ (Fig. S1A)

Click here for additional data file.

Figure S3 Von Mises stress distribution results when the impact force was applied on the superior aspect of the spermaceti junk (Fig. S1B)

Click here for additional data file.

Figure S4 Von Mises stress distribution results when the impact force was applied on the middle aspect of the spermaceti junk (Fig. S1C)

Click here for additional data file.

Figure S5 Von Mises stress distribution results when the impact force was applied on the inferior aspect of the spermaceti junk (Fig. S1D)

Click here for additional data file.

Figure S6 Von Mises stress distribution results when the impact force was applied on the spermaceti junk (Fig. 3) with an acute angle of 20.56°

Click here for additional data file.

Figure S7 Von Mises stress distribution results between Model A with simplified partitions, Model A2 with partition thickness as per Clarke, 1978, and Model C with no partitions

Click here for additional data file.

Figure S8 Von Mises stress distribution results between Models A, B and Model C with twelve, six and zero connective tissue partitions and similar system stiffness

Click here for additional data file.

Table S1 Magnitude of the force assigned to all FEMs to evaluate the sensitivity of the skull stresses on variations of the topological application of the impact force (Fig. S1)

Click here for additional data file.

Supplemental Information 4 Python script used for the probabilistic simulation of all finite element models

Click here for additional data file.

We thank Sandra Lanham for sharing with us her report on the sperm whale ramming incident that occurred near Mildriff Island at the Gulf of California, on the 30th of January 1997. We are also grateful to Dr. Ali Nabavizadeh (University of Chicago) for creating Fig. 1. Our most sincere thanks go to our editor and the two reviewers for the very constructive comments and suggestions during the review process that improved the quality of our work.

Additional Information and Declarations

Competing Interests

Author Contributions

Data Availability

Panagiotis Spyridis is an employee of Polytropos Ltd, London.

Olga Panagiotopoulou conceived and designed the experiments, performed the experiments, analyzed the data, wrote the paper, prepared figures and/or tables, reviewed drafts of the paper.

Panagiotis Spyridis conceived and designed the experiments, performed the experiments, analyzed the data, reviewed drafts of the paper.

Hyab Mehari Abraha analyzed the data, reviewed drafts of the paper.

David R. Carrier conceived and designed the experiments, reviewed drafts of the paper.

Todd C. Pataky performed the experiments, analyzed the data, contributed reagents/ materials/analysis tools, reviewed drafts of the paper.

The following information was supplied regarding data availability:

Dryad DOI: 10.5061/dryad.81rp6.

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
