# Peer review of "Architecture of the sperm whale forehead facilitates ramming combat"

_PeerJ, doi:10.7717/peerj.1895_

## Round 0.1 · original submission · Major Revisions

This manuscript is a wonderful example of how computer modeling can be used to explore functional hypotheses for structures that are difficult to access in vivo. The overall comments from the reviewers are fairly positive, but some additional modeling runs and considerations of the overall anatomy should be added in revision. I have marked this as "major revisions" mainly because of the additional models required; my hope is that the overall content of the manuscript will not require major changes, beyond the points raised by the reviewers.

1) Reviewer 2 notes that the mere presence of partitions with different material properties will affect the model. Thus, the reviewer recommends separating discussion of the function for the overall structure from the function for the materials within the structure. This is a subtle difference, but one that could and should be explored with a few sentences, at least.

2) Reviewer 2 also notes that the shape of the partition differs in the model from reality (uniform thickness vs. tapering). This undoubtedly would affect the results, and should be incorporated into a revised model.

3) Reviewer Huggenberger notes that the spermaceti organ would be a likely point of initial contact during "head-butting," a small contrast with the model presented here. Although my reading of the paper suggests that you effectively did include the spermaceti organ, I do think that impacts against this region should be modeled. I suspect you would likely get similar results, but at the least it would be (potentially) more biologically realistic. This is a very relevant omission from the model, and should be addressed.

4) Following on point 3 above, I would recommend additional simulations with the load applied at different locations on the front of the head or at different angles. At least in terrestrial organisms that head butt, there is a fair degree of variation in location of impact. Given the paucity of data on where the impact occurs in whales, I think it is important to test the role of the junk in all of this and further bolster your hypothesis. I would recommend running the model with a few different angles, smaller areas of force application, etc. This would provide a much more robust investigation and a more convincing exploration of the hypothesis.

5) In lines 106-107: "it is the junk, not the spermaceti organ, that has been suggested to function as a battering ram during aggressive encounters" -- What is this based on? Functional inference? Observations? The observation of forehead-to-forehead ramming is important, but I agree with the reviewer that the spermaceti organ region may be just as relevant as a point of contact.

·

Basic reporting

Dear Editors,
Dear Authors,

The manuscript

“Architecture of the sperm whale forehead facilitates ramming combat”

by Olga Panagiotopoulou, Panagiotis Spyridis, Hyab Mehari Abraha, David R Carrier, and Todd Pataky

meets all the editoral criteria of PeerJ and is a well written scientific paper. However, there is a major week point in the design of the study regarding the anatomy of the sperm whale model used in FEA. The authors test the impact during a ramming event on the junk of the sperm whale nose. However, the junk is not the most rostral part of the nose in adult males. The rostral tip of the spermaceti organ (SO) is situated more rostrally than the junk (see Fig. 1 and Cranford 1999). Thus, the SO must be hit first during “head-butting” and not the junk. I suggest that the authors should either run the same tests with a model of the sperm whale nose that includes the SO (and I would be very interested to see these results) or explain in detail why the SO should not be involved in potential ramming events although it is the most rostral tip of the head of adult males.

With kind regards,
Stefan Huggenberger


Further minor suggestions:

ABSTRACT
What is the tip of the skull (anatomic structure)? Do you mean the rostrum?
Please report important and concrete results in the abstract

DISCUSSION
3rd paragraph: The main function of the junk may be focusing of sound signals used in echolocation and should be less important for the emission of social signals.

Experimental design

see above

Validity of the findings

see above

Additional comments

see above

Reviewer 2 ·

Basic reporting

The authors presented an interesting study about an impact-protect system in the forehead of the sperm whale. The major organs like connective tissue partitions, spermaceti organ and skull were abstracted and a FEM model was established assuming all materials were elastic. The influence of the variation of the elastic modulus was considered using Monte Carlo simulation. And the author found the stress in the model with more partitions was reduced.

Experimental design

1. From an engineering point of view, the model with more partitions uses more material with higher elastic modulus, so it is obvious that it has higher stiffness and lower stress response. The function of this architecture should be discussed separated from the function of the material.
2. As mentioned in the paper, the partition becomes thinner towards the posterior. However, in the FEM model, the partitions and the intervals are uniform.
3. The influence of the variations of the parameters can be directly discussed by adopting different values in the simulation instead of Monte Carlo simulation.

Validity of the findings

The effect of reducing the stress should be discussed more detail. The resistances of different organs are different. For the connective tissue, the reduction of stress may be small compared to its hardness.

Additional comments

1. In 466~467, legends of the Fig.2. The Young’s moduli of the skull should be 14.8 GPa.
2. In 238, a line break is missing.

Annotated reviews are not available for download in order to protect the identity of reviewers who chose to remain anonymous.

---

## Round 0.2 · Minor Revisions

The manuscript is nearly ready for acceptance; please respond to the final minor comment from the reviewer, and then the paper should be good to go.

·

Basic reporting

Dear Editor,
Dear Authors,

The first revision improved the manuscript of Panagiotopoulou et al. "Architecture of the sperm whale forehead facilitates ramming combat" significantly and I would be happy to see it published. It was already obvious from the first reactions after the paper was published as preview on the PeerJ web page that it attracted some attention in the community. Thus, I am sure this paper will stimulate the scientific discussion on this fascinating topic of cetology.

Just a very minor suggestion:
The term “spermaceti junk” was not used in recent papers on this topic and not in the Introduction. It may create some confusion because there is the dorsal “spermaceti organ”, on the one hand, and the “junk”, in the other hand. One may have the impression that the authors could mean both with the term “spermaceti junk”. This potential source of confusion should be avoided.

Sincerely yours,
Stefan Huggenberger

Experimental design

No Comments

Validity of the findings

No Comments

Additional comments

see above Basic Reporting

Reviewer 2 ·

Basic reporting

No comments

Experimental design

No comments

Validity of the findings

No comments

Additional comments

The author gave reliable responses to the concerns and added several simulation results.

---

## Round 0.3 · accepted · Accept

The overall content changes have been completed satisfactorily, following comments from the reviewers. In my final review of the manuscript, I noted a few minor typos or clarifications; these can be corrected in proof, in my opinion.

Abstract: "scaring" should be "scarring"
Line 75: For "ondocete," do you mean "odontocete"?
Line 326: remove comma after "partitions"
Line 333: "Models" should be lower-case
Line 366: "scaring" should be "scarring"
Line 390: Is "developmentally" more appropriate? (or maybe not?)
Line 396: "the" in "the Essex" doesn't need to be italicized
Line 400: "Bovidae" should be capitalized - or write as "bovids"; also "Cetacea" or "cetaceans"
Line 413: "in which head-butting" (add "in")